# ENVIRONMENTAL AI RESEARCH PRIORITIES: WHAT THEY REVEAL ABOUT OPTIMISM AND MISALIGNMENT

## ABSTRACT

While artificial intelligence (AI) is increasingly integrated into environmental research, a comprehensive evaluation of this integration remains limited. In response to this, we analyzed a focused sample of 106 publications from *Nature* and *Science* (2017–2024) to characterize how AI's role in addressing the environmental crisis is represented within these journals. Our mapping reveals a substantial imbalance across all publication types in our dataset, with 73.6% of the publications focusing on Forecasting, 19.8% on Monitoring & Assessment, and only 6.6% on Mitigation. Notably, 81.1% reference prior non-AI approaches, indicating that AI is often used for already-addressed environmental challenges. Most studies rely on standard machine learning techniques and remain at early development stages. Optimism about AI's potential has increased over time; however, high novelty AI uses remain exploratory and rarely operational. These findings highlight trends in how AI is portrayed, deployed, and aligned with environmental priorities, and the importance of reflecting on their implications.

## 1 INTRODUCTION

With the dramatic consequences of the environmental crisis increasingly felt globally through extreme weather events, biodiversity loss, and ecosystem disruption, tackling them has emerged as an unprecedented social challenge that defines our time Esperon-Rodriguez et al. (2022); Newman & Noy (2023); Forster et al. (2024). Addressing this crisis involves both understanding complex environmental systems and their future trajectories, and developing responses to anticipated changes that span technological and social dimensions Clarke et al. (2022); Morris et al. (2025); Dietz et al. (2021); Wunderling et al. (2024). This involves challenges ranging from reducing greenhouse gas emissions through technological and social changes Nelson & Allwood (2021); Probst et al. (2021); IEA (2023), to building resilience to environmental risks through adaptive measures such as flood management systems and coastal protection Bloemen et al. (2018), to forecasting environmental changes such as extreme weather patterns and climate impacts to inform decision-making, and, further, to fostering the social transformations necessary to support sustainable practices. The scale and urgency of these interconnected challenges represent an unprecedented test for human societies.

Artificial intelligence (AI) represents another unprecedented challenge. The concept of AI seems to encompass several technologies, leaving it open what exactly qualifies as artificial intelligence. Yet despite this ambiguity, AI has been increasingly presented as a promising solution to address environmental challenges. AI's potential is connected to its ability to be used to process and analyze large amounts of data using machine learning and deep learning, resulting in enhanced analytical capabilities. To start with, for AI to be beneficial in addressing the environmental crisis rather than exacerbating it, emissions reductions from its use must surpass its carbon footprint Mytton & Ashtine (2022); Masanet et al. (2020). For now, the environmental footprint of AI is drawing attention Bogmans et al. (2025), in fact, the energy consumption of AI systems, particularly those utilizing generative AI, has prompted some technology companies to revise their environmental targets Microsoft (2025). AI is promoted as essential for environmental action beyond the issue concerning its own environmental footprint. However, comprehensive evaluation of how AI is actually being used in environmental research remains limited. To date, most research on AI and the environmental

crisis has focused on assessing the potential of AI to help address versus exacerbate environmental problems Konya & Nematzadeh (2024); Somoye & Akinwande (2025); Rolnick et al. (2022).

Both the scientific community and initiatives by international organizations, such as the United Nations Environment Programme and the World Economic Forum, have advocated for a cautious approach to AI and for incentivizing use cases that align with global environmental ambitions. An important study by Rolnick et al. (2022) has reviewed key areas where machine learning can be used to accelerate the energy transition and mitigate the effects of the environmental crisis, detailing uses where machine learning is expected to have a particularly high impact, while highlighting uncertainties and limitations Rolnick et al. (2022). Konya and Nematzadeh Konya & Nematzadeh (2024) have further mapped where AI is being used in environmental issues, but do not engage the discursive or epistemic implications of such uses. Certain studies have aimed to understand AI's environmental awareness, particularly in the case of generative AI, largely finding that AI is not yet at a stage where models can reliably optimize for sustainability goals Vartziotis et al. (2024; 2025); Strubell et al. (2020); Atkins et al. (2024). These studies primarily focus on quantifying AI's technical capacity to help address the environmental crisis, without examining the patterns of how AI is actually being used—whether research directions are driven by computational feasibility or environmental urgency, or how the portrayal of AI in scientific discourse of relevance to the environmental crisis might influence research priorities and public expectations. Here we analyze a targeted corpus from Nature and Science (2017–2024) to characterize (i) the tasks AI addresses, (ii) development stages, and (iii) how optimism and limitations are represented.

A small but growing body of social science studies has begun to examine how artificial intelligence is generally presented and legitimized in scientific and environmental discourse. For example, Munk et al. Munk et al. (2024) analyze how AI is consistently presented as a problem solving tool in scientific abstracts, while Brevini Brevini (2020) critiques the mythologizing of AI in public and policy narratives that obscure its material and environmental costs. Nalau Nalau (2024) raises concerns about the assumptions embedded in AI-driven decision-making in adaptation science, while Gundersen et al. Gundersen et al. (2022) highlight how data-driven models and technocratic approaches that accompany them can obscure uncertainty and marginalize alternative perspectives in environmental policy. From a philosophical standpoint, Coeckelbergh Coeckelbergh warns that algorithmic logics in environmental governance may be narrowing political imagination.

Building on these studies, we employed an interdisciplinary approach to analyze the use of AI in environmental research between 2017 and 2024, a critical decade during which both AI capabilities and environmental urgency gained unprecedented attention. Through structured search and screening procedures (see Appendix, S0.1), we identified 106 publications that substantively engage with both AI systems and environmental problem solving. These publications include Research Articles, Reviews, Editorials, News, Comments & Opinions, Advertisements, Surveys, and Podcasts, reflecting the diversity of scientific communication in *Nature* and *Science*. This forms a focused sample of high-impact content published in *Nature* and *Science*, two of the most influential interdisciplinary scientific journals. Being also leaders in hosting news about state-of-the-art research, these journals are widely read, cited, and largely trusted by the scientific community, while also receiving the attention of a broader public Baldwin (2015). As such, they offered a suitable dataset for our analysis. Our dataset, comprising 106 publications, addresses environmental issues ranging, indicatively, from weather forecasting to biodiversity assessment to emissions management. We conducted a structured analysis of each publication using a structured framework that captured the task type (forecasting, monitoring and assessment, or mitigation), the environmental issue addressed, the AI development stage, the level of novelty of AI, stated limitations with respect to AI use, as well as the level of optimism related to AI. Our goal was to map where AI is being used in environmental research and to examine how it is portrayed, with particular attention to the assumptions, omissions, and narrative frameworks that shape its representation in relation to environmental challenges and expectations.

The following three research objectives guided this study. First, we examined how AI is presented in addressing environmental issues, assessing whether key factors such as technological limitations, feasibility constraints, and the stage of AI development are acknowledged, as well as whether AI's carbon footprint and other limitations are discussed. Second, we examined the environmental issues addressed using AI, the AI approaches employed, and the patterns of use across various environmental areas. We analyze what is presented as the contributions of AI to environmental problem solving and assess the extent of adoption of AI in various environmental challenges, highlighting

trends, gaps, and limitations in its use. Third, we examine the trends in discourse about AI use in environmental research over time, with the aim of highlighting how the perceived role of AI in environmental discussions has evolved, identifying changes in expectations, uses, and significance. This includes examining changes in AI-related optimism and pessimism across different environmental issues and periods. Additionally, we assessed how the terminology and presentation of AI in scientific discourse have changed, reflecting broader shifts in the perception of AI's potential in addressing environmental issues.

Our approach examined how these two complex and evolving challenges—the environmental crisis and the development of AI as both a potential solution and a challenge in its own right—intersect in scientific discourse during a critical decade in which both gained heightened attention and visibility. Our results reveal key trends in task types, development stages, and portrayals of this promising yet complicated technology as it is positioned in relation to contemporary environmental research.

## 2 METHODS

We performed a structured analysis of how artificial intelligence (AI) is positioned in addressing environmental challenges. The corpus comprises publications from *Nature* and *Science*, two journals chosen for their interdisciplinary scope and high influence on both academic and public discourse. The review spanned the years 2017 to 2024 to capture developments in AI uses addressing environmental issues over the past decade. This period follows key moments such as the adoption of the Paris Agreement in 2015 United Nations Framework Convention on Climate Change (UNFCCC) (2015), which helped shape global research agendas. It also reflects the rapid growth of AI research, particularly in machine learning and neural networks LeCun et al. (2015), which increasingly informed environmental monitoring and sustainability initiatives Reichstein et al. (2019).

### 2.1 SEARCH STRATEGY AND INCLUSION CRITERIA

We searched the Nature and Science websites using predefined keyword combinations (see SI for full logic and dates). Because the journal search interfaces update dynamically, the initial hits numbered in the several hundreds across both sites and exact per-journal counts were not logged. From the first reproducible screening step onward we recorded counts as follows. First, we screened titles and, when available, abstracts to identify items substantively engaging both artificial intelligence and environmental issues, retaining 113 records. Second, to avoid double-counting, we removed overlapping coverage where a news item reported on a research article already in the set (2 removed; 111 remaining). Third, we assessed full texts for eligibility and excluded 5 items that discussed AI but did not substantively address the environmental crisis, yielding a final corpus of 106 publications. A study-selection overview appears in Fig. A1, detailed search terms and dates are provided in the Appendix.

### 2.2 ANALYTICAL FRAMEWORK

To guide our analysis, we developed an interdisciplinary framework of 18 questions (Table 1), grounded in insights from science and technology studies (STS), environmental science, and computer science. Each publication in our sample was analyzed according to this framework, which captures both technical and discursive aspects of AI use in environmental research. The questions are organized into five analytic dimensions:

1. **Portrayal and Language**: Assessing tone, optimism or pessimism, emotionally charged language, and whether AI is portrayed as contributing positively or negatively to environmental outcomes.

2. **Research Focus and Development Stage**: dentifying if the focus is on model development or problem-solving. Systems were classified into three maturity levels:: **(i)** Research: Methods tested in simulated environments on benchmark data, used only by the researchers; **(ii)** Demonstration: Functional prototypes tested in a controlled setting on real-world data, used by authors or internal testers; and **(iii)** Deployment: Methods integrated into a live or real-world operational scenario.

3. **Limitations and Feasibility Constraints**: Capturing mentions of data, model and other limitations, and carbon footprint

4. **AI Systems and Novelty**: Classifying the type and originality of AI systems (e.g., machine learning, deep learning, reinforcement learning). We assessed the type and originality using a single-dimensional rubric ranging from 1 to 3, which was used to classify papers into three categories: *(1)Highly Novel*: The work introduces a new architecture or defines a new paradigm, supported by rigorous state-of-the-art comparisons; *(2) Moderately Novel*: The work presents an incremental architectural improvement or applies a known method to a new domain in a significant way; *(3) Incremental/Standard*: The work applies a standard, off-the-shelf model to an established problem with weak or absent baselines.

5. **Publication Type**: Categorizing publications by publication types (e.g., research, review, news, editorial)

Descriptions of each question and how responses were classified are included in the Appendix. The responses to these questions formed the basis of our analysis and results. The responses were recorded in a structured dataset with numerical, binary, categorical, and open-ended fields. Each publication was independently reviewed by one author, and a subset was cross-checked by others for consistency (see SI for more info). Any discrepancies were resolved through discussion. The publications were categorized according to the environmental issue addressed and AI's role in environmental problem solving. This inductively developed classification revealed three primary uses of AI: Forecasting, Monitoring & Assessment, and Mitigation. We analyze Monitoring & Assessment jointly. We define monitoring as repeated measurement or mapping of environmental states over time, and assessment as evaluating current status or implications against criteria (e.g., thresholds, targets) at a given time. In our data, Monitoring & Assessment typically co-occur; we therefore analyze them jointly by functional role. For brevity, we occasionally refer to AI's "role in environmental problem solving" as "task type" in figures and tables. These were cross-referenced with environmental subcategories (e.g., air quality, biodiversity) to form a structured matrix (see Table A5). A full breakdown of uses and corresponding references appears in Table A6.

Table 1: Questions for structured analysis of reviewed publications

| No. | Question | Possible responses |
|-----|----------|--------------------|
| **Category I: Portrayal and language** | | |
| Q1 | Does the publication explicitly use the term "Artificial Intelligence"? | Yes/No |
| Q2 | Is the publication optimistic or pessimistic about AI's potential to address the environmental crisis? | Scale: 1 = Optimistic, 5 = Pessimistic |
| Q3 | Is the publication optimistic or pessimistic about the effectiveness of its specific AI model in addressing the environmental crisis? | Scale: 1 = Optimistic, 5 = Pessimistic |
| Q4 | Does the publication use language biased toward either helping or worsening the environmental crisis? | Scale: 1 = Helping, 5 = Worsening |
| Q5 | Does the publication use strong or emotionally charged words to describe the success or failure of the AI model? | Yes/No; examples |
| **Category II: Research focus and development stage** | | |
| Q6 | What does the publication primarily focus on: AI system development or addressing the environmental issue? | Model/Environment/Both |
| Q7 | At what stage of development is the AI system? | Research/Demonstration/Deployment |
| Q8 | Which environmental issue does the publication address? | Flooding, biodiversity, sea level rise, etc. |
| Q9 | Which aspect of environmental problem solving (task type) does the publication focus on? | Forecasting/Monitoring & Assessment/Mitigation |
| **Category III: Limitations and feasibility constraints** | | |
| Q10 | Does the publication mention model limitations? | Yes/No |
| Q11 | Does the publication mention data limitations? | Yes/No |
| Q12 | Does the publication mention other limitations? | Yes/No |
| Q13 | Does the publication mention bad use examples? | Yes/No |
| Q14 | Does the publication mention the carbon footprint of AI? | Yes/No |
| **Category IV: AI systems and novelty** | | |
| Q15 | What type of AI approaches are used in the publication? | e.g., ML, DL, etc. |
| Q16 | How novel is the AI system used in the publication? | Scale: 1 = Very novel, 3 = Not novel, NA |
| Q17 | Does the publication mention whether the environmental issue was previously addressed using other non-AI approaches? | Yes/No |
| **Category V: Publication type** | | |
| Q18 | What type of publication is it? | Research article, review, editorial, comments & opinions (C&O), news, advertisement (Ad), podcast, survey |

## 3 RESULTS

Our structured analysis of 106 publications reveals key trends across four main aspects: how AI is presented in environmental research, how it is used across different environmental tasks, how portrayals have evolved over time, and how it relates to previous non-AI approaches.

### 3.1 TECHNO-OPTIMISTIC BIASES

One of the most consistent themes across the 106 publications is the predominantly optimistic tone with which AI is presented as a solution to environmental challenges. This optimism deepens over time, and the count of highly optimistic items is largest in 2023, while neutral portrayals are most common in 2022 (Figure 1a). The crest in 2022–2023 aligns with heightened visibility of AI research and climate-innovation coverage, suggesting that scholarly narratives may have moved in step with broader techno-policy attention. Although there is a small uptick in critical pieces after 2022, pessimistic accounts remain rare. Authors also tend to be at least as optimistic—and often more so—about the effectiveness of their own systems (see dashed overlays in Fig. 1a) than about AI in general (solid bar lines), a pattern consistent with how novelty and utility are rewarded in academic publishing.

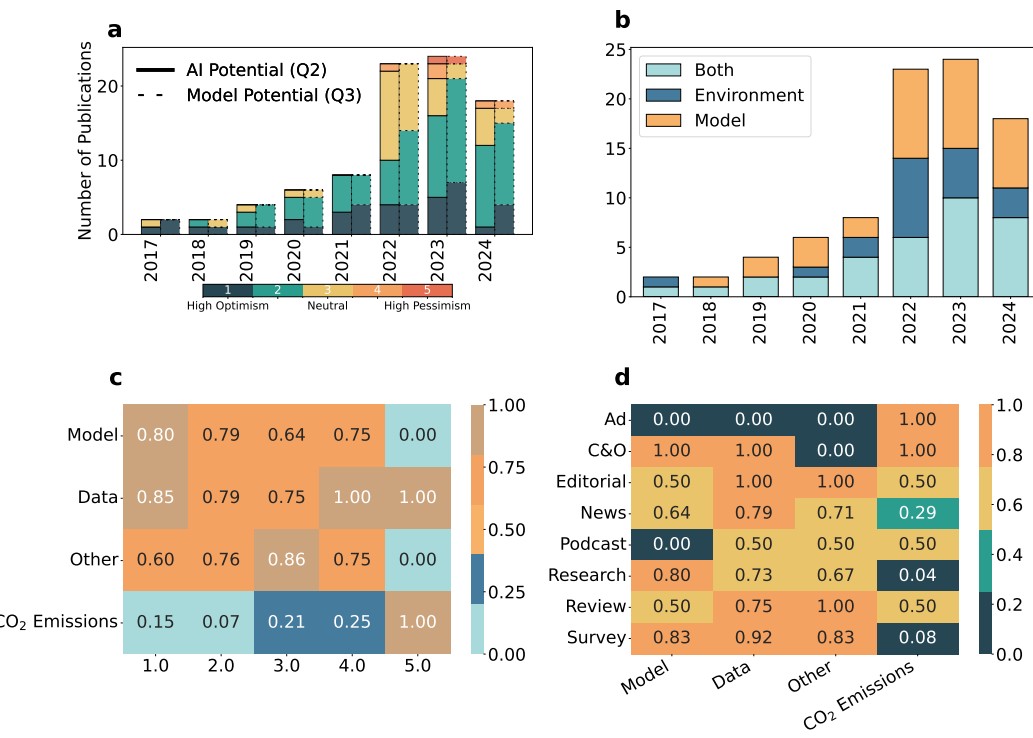

Figure 1: **a** Perceptions of AI's role in environmental research over time (2017–2024), based on two dimensions of optimism and feasibility. Stacked bars show how optimistic each publication is about AI's overall potential to address the environmental crisis (Q2 in Methods). Dashed overlay bars represent optimism about the effectiveness of the specific AI system discussed (Q3). Scores range from 1 (high optimism, dark green) to 5 (high pessimism, red). **b** Primary focus of each publication over time—whether centered on developing the AI model (orange), addressing the environmental problem (navy), or giving equal weight to both (light blue). **c** Proportion of publications mentioning different types of feasibility limitations—model, data, other, and carbon footprint ($CO_2$ emissions)—grouped by optimism level. **d** Proportion of publications mentioning each category of feasibility limitation, broken down by publication type (e.g., Advertisement (Ad), Comments & Opinions (C&O), News, etc.).

The focus of publications shifts over time (Figure 1b). Early work often treats environmental problems chiefly as testbeds for model development. From 2020 onward, more items give balanced space

to both technical advance and environmental goals—this balanced focus peaks in 2023 and remains elevated thereafter. Emphasis on the environment alone peaks in 2022 before receding, indicating that technical advancement continues to anchor the conversation even as application concerns grow.

Limitations are acknowledged unevenly across genres and tones (Figure 1c–d). Model and data limitations are discussed routinely—even in optimistic pieces—while other feasibility constraints (institutional, operational, governance-related) are also common but tend to be articulated more fully in reviews, surveys, and news than in research articles. Carbon footprint appears infrequently overall and shows up across optimistic, neutral, and critical accounts; research articles rarely discuss it (3/70; roughly 1 in 25), whereas broader genres raise it more often. Taken together, these patterns suggest that enthusiasm for technical progress can eclipse downstream feasibility and footprint considerations, especially in primary research formats.

## 3.2 Distribution of AI Tasks Across Environmental Issues

To understand how AI is being mobilized in environmental research, we classified the 106 publications in our dataset into three major problem solving tasks (or task types): forecasting (73.6%), monitoring & assessment (19.8%), and mitigation (6.6%) (see Table A5 in Appendix). Forecasting task type clearly dominates the field. These three task types span 28 distinct environmental issues, with weather forecasting (24 publications), biodiversity assessment (12), and climate model forecasting (10) among the most common (see Table A6 in Appendix). This dominance of forecasting likely reflects the compatibility between structured environmental data (e.g., weather records, climate simulations) and widely used machine learning techniques. In contrast, monitoring & assessment and mitigation tasks often rely on more heterogeneous or dynamic inputs.

Chronologically, forecasting appeared earlier, with the first *Nature* publication in 2017 and the first *Science* publication in 2018. Monitoring & assessment followed (appearing in *Nature* in 2017 and *Science* in 2021), while the first mitigation pieces appeared in *Nature* in 2018, and not until 2022 in *Science*. Since then, forecasting uses have steadily grown, while monitoring & assessment tasks occupy a smaller, while monitoring & assessment tasks occupy a smaller, though consistent, share of applications. Mitigation, by contrast, emerged later and remains sparse throughout, represented by sporadic contributions. Together, these patterns suggest that AI development has so far followed established data infrastructures more readily than direct climate intervention needs.

## 3.3 Novelty and Development Stage

While the imbalance across task types reflects how different environmental challenges align with AI's current strengths, a separate pattern emerges when examining the novelty of the AI systems themselves. As shown in Figure 2 and detailed in Table A9 (see SI), nearly half of the publications rely on moderately novel AI systems (score=2), while highly novel approaches (score=1) appear in only 20.75% of cases. This reflects a general preference for established AI systems over experimental ones. Novelty distributions are comparable across groups: the mean novelty score is 2.10 for research articles and 2.03 for other publication types (Table A1). Notably, mitigation exhibits the highest average novelty (1.86), despite representing only a small subset of the dataset (Table A10 in Appendix). This suggests that experimentation is more common in unsettled domains where modeling frameworks and evaluative standards are lacking, whereas monitoring and assessment tasks tend to rely on structured indicators and established techniques, leading to more standardized approaches.

Across task types, most uses remain at the research or demonstration development stage, with limited real-world deployment (Figure 2). Even highly novel approaches—more common in publications published after 2022—are largely confined to academic or pilot contexts. This combination of methodological conservatism and limited implementation reinforces a landscape where innovation remains exploratory, despite frequent claims of transformative potential.

## 3.4 Sensitivity to Publication Type

In order to ensure that our findings are not influenced by the genre mixing within *Nature* and *Science*, we compared the results separately for Research Articles (RAs) and all other publication types. Of the 106 publications retained, 70 were classified as Research Articles and 36 as other publication types (Reviews, News, Editorials, etc.). A research-articles-only view yields similar optimism

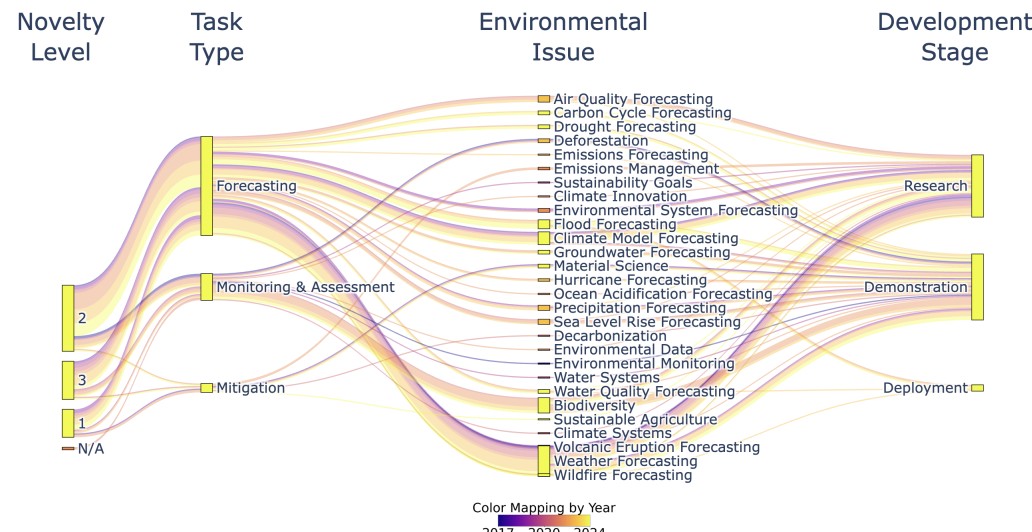

Figure 2: The flow of AI uses in environmental research across four dimensions: novelty level, role in environmental problem solving (task type), environmental issue addressed, and development stage. Each node represents a category, with node size indicating the number of publications in that category. Each line represents one publication flowing through the corresponding nodes across all four dimensions. Node colors indicate the average publication year for publications in each category, while line colors correspond to individual publication years (darker = earlier publication year). This visualization reveals how AI uses of varying novelty levels are distributed across different task types, environmental issues, and development stages, and how these trends have evolved over time.

patterns: overall-potential optimism (Q2) averages 2.22 for research articles vs. 2.17 for other publication types, and model-specific optimism (Q3) averages 1.90 vs. 2.14 (among non-missing; see Tables A2–A3 in Appendix). When limited to research articles, the corpus becomes even more forecasting-dominant (56/70; 80%), with mitigation uses notably underrepresented (2/70; 2.9%) (see Table A4 in Appendix). Novelty distributions are comparable across groups (mean 2.10 for research articles vs. 2.03 for other publication types; see Table A1 in Appendix).

### 3.5 REFERENCING OF NON-AI APPROACHES

Of the 106 publications analyzed, 86 (81.1%) explicitly reference prior non-AI approaches to the same environmental problem (see Table A7 in the SI). This pattern holds across tasks, with some variation by task type and environmental subcategory. As shown in Figure 3 and Table A8 (see SI), forecasting publications almost universally acknowledge prior approaches—several environmental issues show complete continuity—so AI is largely presented as improving established modeling workflows rather than solving previously intractable problems. Mitigation, although smaller in volume, also typically situates AI alongside existing tools and policies. By contrast, Monitoring & Assessment shows greater variability: references to prior approaches are comparatively sparse, and some items present AI-enabled analyses without explicitly linking to a method lineage. A similar pattern appears across publication types. Research articles most often acknowledge prior approaches, while a minority do not; the remaining non-mentions are spread across reviews, surveys, and news in small numbers. Overall, environmental AI is primarily positioned as building on established solution spaces—promising gains in accuracy, scale, or speed—rather than opening wholly new problem domains.

## 4 DISCUSSION

Our analysis reveals a mismatch between AI research priorities and environmental urgency. While AI is increasingly portrayed as essential for environmental action, the overwhelming focus on prediction (73.6% of uses) over mitigation (6.6%) is consistent with research activity tracking com-

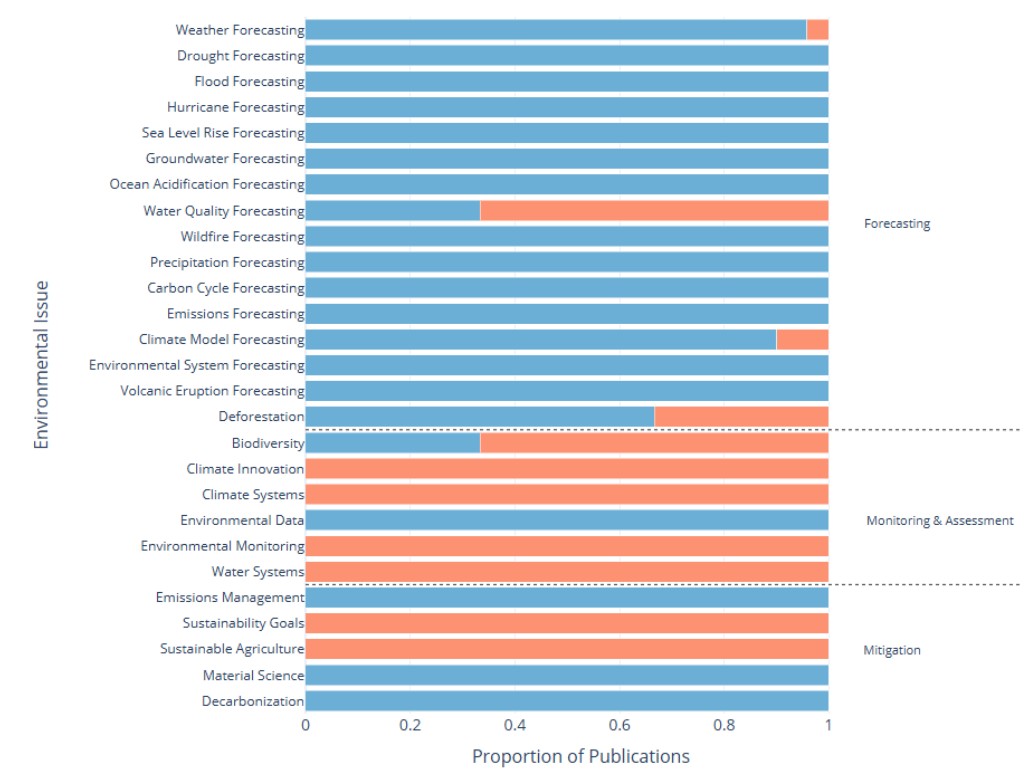

Figure 3: Proportion of publications, by environmental issue, that indicate whether the issue had previously been addressed using another non-AI approach. Bars are color-coded by response: Yes (blue), No (soft red), and Not Available or Not Specified (gray). Environmental issues are grouped by AI task type (dashed lines).

putationally tractable problems more than direct intervention needs in this corpus. This imbalance exposes a fundamental tension in research priorities. Forecasting tasks dominate—weather forecasting alone accounts for 24 of 106 uses. We observe that AI tends to produce better prediction tools but contribute little to concrete environmental interventions. Another reason for this is because the current AI tools can't perform what-if tasks easily. This challenge is reflected in the lower, more novel scores observed in mitigation uses (1.86 versus 2.08 for forecasting), suggesting a form of epistemic narrowing where problems are prioritized by computational tractability rather than environmental significance. Furthermore, this results in fewer deployment-stage applications relative to research or demonstrations, leaving operational interventions underrepresented in this corpus.

Additionally, we observe fewer deployment-stage applications relative to research or demonstrations (Fig. 2), so contributions to operational interventions are less frequently represented in this corpus. Beyond deployment gaps, the relationship between optimism and acknowledgment of limitations reveals notable blind spots in the way AI's environmental potential is portrayed. Although technical limitations receive widespread attention—appearing in approximately 80% of publications—the carbon footprint of AI is mentioned only in 14 publications. Carbon footprint is rare in research articles (3/70; 1 in 25), with higher mention rates in other formats (Fig. 1c-d; Table A7). Footprint appears across all three task types (forecasting, monitoring & assessment, mitigation), but not across every specific environmental issue. The publication type analysis reveals that these blind spots are institutionally embedded. Research publications—while the majority—focus heavily on technical performance and rarely discuss carbon footprint (∼4%). These omissions aren't from lack of awareness; other formats engage more directly with broader feasibility. This trend suggests that publication norms and incentive structures may shape how AI's environmental potential is represented, but we do not identify mechanisms with these data.

The temporal evolution of optimism further highlights these dynamics. The peak in positive framing occurs in 2023, following the build-up in 2022, coinciding with both an increase in publication volume and heightened public attention to the environmental potential of AI. The modest increase in more critical perspectives after 2022 may reflect a growing recognition of feasibility constraints and environmental consequences. However, these perspectives remain underrepresented in a field largely shaped by technological optimism and narrowly portrayed feasibility concerns. Perhaps most concerning is the concentration of uses in the early stages of development despite a decade of research investment. Even forecasting tasks, which represent the most mature uses, remain largely at research or demonstration stages. This persistent gap between innovation and implementation suggests that current approaches may be fundamentally misaligned with the practical requirements of environmental problem solving.

The inverse relationship between novelty and deployment is particularly revealing. High-novelty approaches (score=1) appear throughout the timeline and become more frequent after 2022, but remain confined to research contexts. Meanwhile, the moderate novelty approaches (score=2) that dominate the field represent incremental advances rather than transformative capabilities. This suggests that environmental AI research operates within a "comfort zone" of familiar techniques that can be published but not necessarily deployed. The limited mention of bad use examples (19.8% of publications) further indicates selective reporting that may obscure the true challenges of translating computational innovations into environmental practice. When combined with the under-reporting of carbon footprint considerations, this may result in a structural overestimation of AI's environmental benefits.

These findings have direct implications for environmental policy and research funding. The current task distribution suggests that AI resources are being allocated based more on computational opportunity rather than environmental impact. While improved prediction capabilities have important value, the roughly eleven-to-one ratio between forecasting and mitigation uses indicates a possible misalignment between computational opportunity and environmental urgency during a period when environmental scientists emphasize the urgent need for emissions reductions and interventions to reduce the health, economic, and societal impacts of extreme weather events. Moreover, the strong acknowledgment of prior non-AI approaches (81.1% of publications) indicates that AI is primarily positioned as strengthening established workflows—promising gains in accuracy, scale, and speed—rather than opening entirely new problem domains. This incremental progress is valuable, but given the scale and urgency of environmental challenges, it remains limited on its own. Consistent with this, most uses stay at the research or demonstration stage across domains, with relatively few deployments, suggesting that current evaluation and publication incentives may privilege technical novelty over practical deployment, and thereby sustain a research-implementation gap. Consequently, for policymakers and funding agencies, our results suggest the need for deliberate intervention to encourage more allocation of computational resources to mitigation uses. This may require accepting lower tractability in exchange for greater environmental relevance, potentially through funding mechanisms that explicitly reward deployment over publication and environmental impact over technical innovation.

Our analysis reflects the perspectives and priorities portrayed in high-impact publications, which may not fully represent the diversity of environmental AI uses or the full spectrum of research communities working in this area. The focus on the journals *Nature* and *Science*, while providing access to high-impact research, may underrepresent work from applied research contexts, developing countries, and interdisciplinary collaborations that operate outside traditional academic publishing structures. Our assessment introduces interpretive elements that, despite cross-validation efforts, may reflect our disciplinary backgrounds and analytical frameworks. However, the consistency of patterns across multiple dimensions of analysis, from use distribution to temporal trends to limitation acknowledgment, suggests that our core findings reflect genuine characteristics of the field rather than methodological artifacts.

## CODE AND DATA AVAILABILITY

The dataset generated and analyzed during this study, along with the code used to produce the figures, are available in an anonymized open-access repository.

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

## A APPENDIX: EXTENDED METHODS

### A.1 LITERATURE SEARCH AND JOURNAL RATIONALE

We conducted a comprehensive review of publications from the interdisciplinary journals *Nature* and *Science*. These journals were selected not because they encompass the full range of environmental AI research, but because of their reputation for disseminating high-impact research and commentary across disciplines. While most environmental AI applications initially appear in specialized environmental science or AI journals, *Nature* and *Science* provide a broader forum where emerging technologies and global environmental challenges are framed and debated, influencing both academic and public discourse. Their emphasis on high-impact research, however, may introduce a bias toward widely recognized technological advancements, potentially overlooking more specialized or domain-specific discussions.

### A.2 SEARCH STRATEGY AND INCLUSION CRITERIA

To identify relevant publications, we applied a set of predefined search terms to the *Nature* and *Science* archives. These included *Climate Change and Machine Learning*, *Climate Change and Deep Learning*, *Climate Change and Neural Networks*, *Climate Change and AI*, *Environmental Crisis and Machine Learning*, *Environmental Crisis and Deep Learning*, *Environmental Crisis and Neural Networks*, and *Environmental Crisis and AI*. Searches were conducted between May and June 2024. These terms were selected to capture broad framings of how AI is positioned in relation to climate change and the environmental crisis, reflecting the interdisciplinary and public-facing language often used in these journals. While "environmental crisis" is not a standard technical keyword, it was deliberately included to capture the broader framing commonly used in *Nature* and *Science* news items and editorials.

An initial search of the *Nature* and *Science* websites using our predefined terms (May–June 2024) returned several hundred hits across both journal sites. Because these web interfaces update results dynamically, exact per-journal hit counts were not logged. From the first reproducible screening step onward, we tracked counts as follows:

1. **Title (and abstract when available) triage:** Retained records that substantively engaged both artificial intelligence and environmental issues (n = 113).

2. **Overlap removal:** To avoid double counting of the same underlying study, we removed news items that covered a research article already retained (n removed = 2; n remaining = 111).

3. **Full-text eligibility:** Excluded **5** items that discussed AI but did not substantively address the environmental crisis, yielding the final corpus (n = 106, publications dated 2017–2024).

Figure A1 summarizes these steps and counts.

### A.3 QUESTION CATEGORIZATION FRAMEWORK

To guide our analysis, we developed an interdisciplinary framework of 18 questions (Table 1). Each publication in our sample was analyzed according to this framework, which captures both technical and discursive aspects of AI use in environmental research. The framework was applied consistently across all publication types, and we found that its questions were applicable beyond research articles. Non-research publications (e.g., news items, editorials, or commentaries) frequently contained substantive information about AI systems, their limitations, or their implications, and were therefore coded when such content was present. When such details were absent, responses were recorded as NA. The questions guiding the analysis were classified into five analytic dimensions, each focusing on a different aspect of AI's role and presentation in environmental discourse. Below, we briefly explain how each question was interpreted during the review process.

**Category I: Portrayal and Language** This category captures how AI is rhetorically portrayed in relation to environmental challenges. Question Q1 records whether the publication explicitly uses the term *artificial intelligence*, helping distinguish technical engagement from broader thematic references. Questions Q2 and Q3 assess how optimistic or pessimistic the publication is—first,

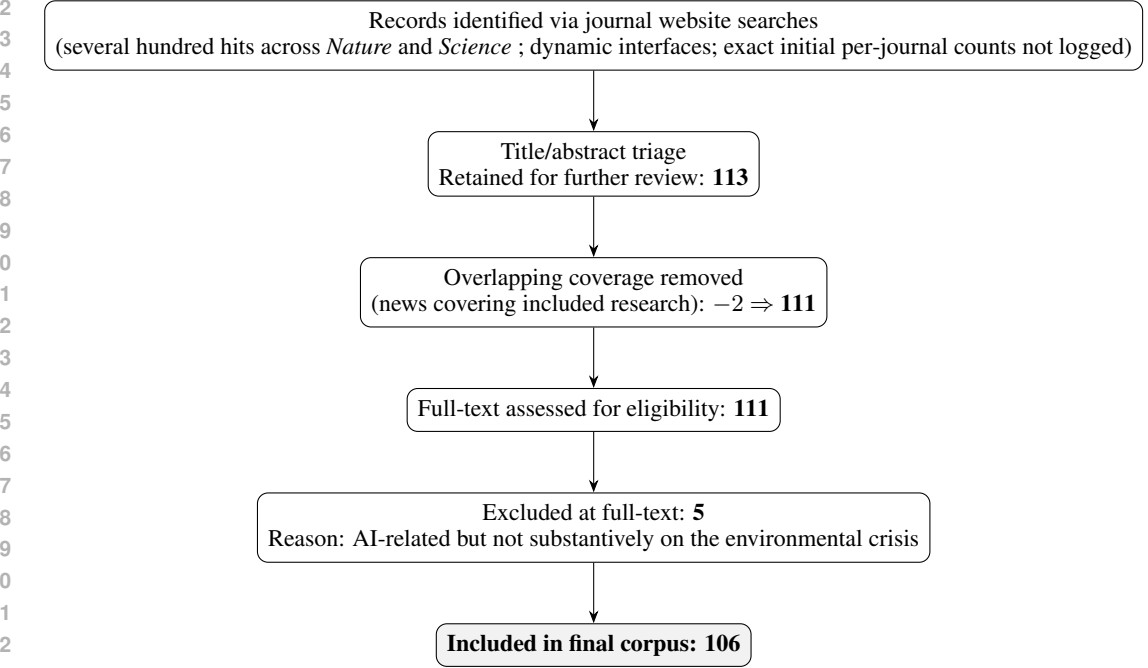

Figure A1: Study selection overview for the *Nature* and *Science* corpus (2017–2024). Counts are shown for each screening step; see Extended Methods for search terms and inclusion criteria.

regarding AI in general as a solution to the environmental crisis, and second, regarding the specific AI model presented in the publication. Question Q4 evaluates whether the language used is biased toward portraying AI as either helping to solve or worsening environmental challenges. Question Q5 identifies the presence of emotionally charged or exaggerated language, particularly where strong or over-optimistic wording is used to describe AI's potential or capabilities.

**Category II: Research Focus and Development Stage Category II: Research Focus and Development Stage** This category examines the publication's primary focus and the stage of development of the AI use. For Question Q6, publications were categorized by their primary focus as model, environment, or both. "Model" publications emphasized AI system development, often with limited attention to environmental implications. "Environment" publications focused primarily on addressing environmental challenges, where AI served as a tool rather than the main object of innovation. The "Both" category was used when publications engaged substantively with both AI model development and addressing the environmental issue, giving equal weight to technical and environmental considerations. When classifying borderline cases, we coded each publication according to its dominant emphasis: "Model" when technical development or algorithmic improvement clearly dominated; "Environment" when environmental analysis or application was central; and "Both" when both aspects were substantively integrated or balanced throughout the text.

Question Q7 captures the development stage of the AI system, indicating whether the technology is still in the research phase, undergoing demonstration in case studies, or already deployed in real-world uses. Question Q8 captures the specific environmental issue addressed by the publication, such as biodiversity loss, flooding, or air quality. Question Q9 identifies the aspect of environmental problem-solving (task type) that the publication focuses on—whether forecasting, monitoring & assessment, or mitigation. When multiple task types were discussed, coding reflected the primary functional role of AI within the study—for instance, a work using monitoring data mainly to predict future states was classified as "Forecasting," whereas studies emphasizing situational awareness or evaluation of current conditions were coded as "Monitoring & Assessment."

**Category III: Limitations and Feasibility Constraints** This category assesses whether the publication discusses technical, practical, or ethical challenges related to the use of AI. Question Q10 captures model-related limitations, such as issues with generalization, robustness, or explainability. Question Q11 focuses on data-related limitations, including insufficient data, lack of representa-

tiveness, or biases in training datasets. Question Q12 addresses feasibility constraints not directly tied to models or data—for example, high computational costs, institutional barriers, or difficulties integrating AI into existing workflows. Question Q13 identifies whether the publication mentions risks of misuse or unintended consequences of AI. Question Q14 notes whether the publication discusses the environmental costs of AI itself, such as the carbon footprint associated with training or deploying models.

**Category IV: AI Systems and Novelty** This category identifies the type and originality of the AI systems described in the analyzed publications. Question Q15 records the specific AI systems employed, such as machine learning, deep learning, or hybrid configurations. Question Q16 assesses the novelty of the AI systems, based on whether the publication introduces a new system, adapts an existing one to a new environmental context, or employs a well-established system. Question Q17 notes whether the environmental issue addressed had previously been tackled using non-AI approaches, providing context for how AI is positioned relative to prior approaches. When the relevant information for Questions Q15 or Q16 was not provided or not applicable to the publication type, responses were coded as NA.

**Category V: Publication Type** This final category classifies the type of publication in which the publication appears. Question Q18 records whether the publication is a Research Article, Review, Editorial, Comments & Opinions (C&O), News, Advertisement (Ad), Podcast, or a Survey. This helps contextualize how the content is presented and who the intended audience is for each piece.

The framework was applied consistently across all publication types, and we found that its questions were applicable beyond research articles. Non-research publications (e.g., news items, editorials, or commentaries) frequently contained substantive information about AI systems, their limitations, or their implications, and were therefore coded when such content was present. When such details were absent, responses were recorded as NA. Together, these five categories provide a structured framework for evaluating how AI is portrayed, used, and critically assessed in high-profile environmental research in *Nature* and *Science*.

Operational definitions for maturity levels (Q7) and novelty scoring (Q16) are provided in the Methods section of the main article. For completeness, optimism and pessimism (Q2–Q3) were scored on a five-point scale based on explicit evaluative statements. Specifically, optimism was assessed contextually, based on evaluative language and the strength of claims regarding AI's potential. Statements explicitly portraying AI as transformative or revolutionary (e.g., "spurring a revolution," "game-changer") were coded as highly optimistic (1). Statements that express confidence without exaggeration (e.g., "AI is the right technology for that," "paves the way for successful use") were coded as moderately optimistic (2). Neutral or balanced phrasing acknowledging both potential and limitations was coded as 3, while skeptical or critical statements (e.g., emphasizing risks, uncertainty, or failure) were coded as 4–5. Optimism was therefore evaluated relative to tone and context, rather than the presence of specific keywords alone.

"Other limitations" (Q12) capture feasibility or contextual constraints that go beyond data or model performance. These include institutional or infrastructural barriers (e.g., insufficient monitoring systems, lack of policy support, or limited institutional capacity), epistemic or methodological uncertainties (e.g., challenges in attributing outcomes to AI interventions or in generalizing results across contexts), and operational or ethical concerns (e.g., interpretability, validation requirements, or hesitation to adopt AI systems). "Other" limitations therefore refer to broader systemic or contextual factors that affect the feasibility, credibility, or uptake of AI applications, rather than technical or data-related shortcomings.

## A.4 RESPONSE DOCUMENTATION AND ANALYSIS PROCEDURE

After defining the analytical framework, all responses were systematically recorded and processed. Data cleaning, consistency checks, and visualizations were performed in Python via Google Colab. Systematic categorization allowed for both statistical trend analysis and interpretation. Each publication was reviewed by one coder. A 20% subset (21/106) was independently cross-checked by a second coder to calibrate application of the codebook for interpretive variables (e.g., optimism, novelty); disagreements were resolved by consensus. We did not estimate formal inter-rater reliability statistics and treat these variables as interpretive judgments guided by the codebook.

Table A1: Novelty of AI system by publication group. Values are counts (%) among non-missing; means with 95% confidence interval (CI). Scale: 1=very novel, 3=not novel.

| Novelty level | Research Articles (n=70) | All other types (n=34) |
|---|---|---|
| 1 | 12 (17.1%) | 10 (29.4%) |
| 2 | 39 (55.7%) | 13 (38.2%) |
| 3 | 19 (27.1%) | 11 (32.4%) |
| Mean (95% CI) | 2.10 (1.94, 2.26) | 2.03 (1.76, 2.30) |

*Note:* No missing data for Research Articles (0/70); 2/36 missing for other publication types.

Table A2: Optimism about AI's overall potential (Q2) by publication group. Values are counts (%) among non-missing. Scale: 1=most optimistic, 5=most pessimistic.

| Level | Research Articles (n=60) | All other types (n=35) |
|---|---|---|
| 1 | 10 (16.7%) | 10 (28.6%) |
| 2 | 28 (46.7%) | 14 (40.0%) |
| 3 | 21 (35.0%) | 7 (20.0%) |
| 4 | 1 (1.7%) | 3 (8.6%) |
| 5 | 0 (0.0%) | 1 (2.9%) |
| Mean (95% CI) | 2.22 (2.03, 2.40) | 2.17 (1.83, 2.52) |

*Note:* Missing data for Research Articles (10/70); 1/36 missing for other publication types.

Responses were documented in Google Sheets using dropdown menus for categorical and numerical inputs to ensure consistency. Open-ended responses, such as excerpts or qualitative notes, were recorded manually. The following answer formats were used:

- **Numerical responses**: Assigned for ranked responses (e.g., optimism in Q2 from 1 - 5, novelty in Q16).
- **Yes/No responses**: Used for binary questions (e.g., terminology use in Q1, model limitations in Q10-Q12).
- **Categorical responses**: Applied to fixed-option questions (e.g., AI development stage in Q7, publication type in Q17).
- **Open-ended responses**: Extracted for qualitative detail (e.g., examples in Q5, specific methods in Q15).

### A.5 SUPPLEMENTARY SENSITIVITY TO PUBLICATION TYPE

To assess whether genre mixing influences our findings, we tabulated Novelty and Optimism separately for Research Articles (RAs) and all other publication types (News, Reviews, Surveys/Perspectives, Editorials, Podcasts, Advertisements, Comments & Opinions). Distributions are broadly similar: RAs are slightly more optimistic about their specific model on average, while Novelty remains comparable across groups.

Splitting Research Articles (RAs) from other publication types shows broadly similar novelty and overall-potential optimism patterns; RAs are slightly more optimistic about their own models (Q3), consistent with primary research centering on a specific system. Together with the RA-only task distribution (80% forecasting; 2.9% mitigation), these checks indicate that our main findings are not driven by genre mixing but reflect patterns within research articles as well.

## B APPENDIX: SUPPLEMENTARY RESULTS TABLES

Table A3: Optimism about the specific model (Q3) by publication group. Values are counts (%) among non-missing. Scale: 1=most optimistic, 5=most pessimistic.

| Level | Research Articles (n=68) | All other types (n=29) |
|---|---|---|
| 1 | 18 (26.5%) | 7 (24.1%) |
| 2 | 40 (58.8%) | 13 (44.8%) |
| 3 | 9 (13.2%) | 8 (27.6%) |
| 4 | 1 (1.5%) | 0 (0.0%) |
| 5 | 0 (0.0%) | 1 (3.4%) |
| Mean (95% CI) | 1.90 (1.74, 2.06) | 2.14 (1.80, 2.47) |

*Note:* Missing data for Research Articles (2/70); 7/36 missing for other publication types.

Table A4: Task distribution by publication group. Values are counts (%).

| Group | Forecasting | Monitoring & Assessment | Mitigation |
|---|---|---|---|
| Research Articles (n=70) | 56 (80.0%) | 12 (17.1%) | 2 (2.9%) |
| All other types (n=36) | 22 (61.1%) | 9 (25.0%) | 5 (13.9%) |

Table A5: Classification of publications by AI's role in environmental problem-solving and corresponding environmental issue subcategories

| AI's Role in Environmental Problem-Solving | Environmental Issues |
|---|---|
| Forecasting (78 publications, 16 subcategories) | Weather, Drought, Flood, Hurricane, Sea Level Rise, Groundwater, Ocean Acidification, Water Quality, Wildfire, Precipitation, Air Quality, Carbon Cycle Forecasting, Emissions, Climate Model, Environmental System, Volcanic Eruption and Seismic Events |
| Monitoring & Assessment (21 publications, 8 subcategories) | Deforestation, Biodiversity, Climate Innovation, Climate Systems, Environmental Data Enhancement, Water Systems, Environmental Monitoring, Sustainability Goals, |
| Mitigation (7 publications, 4 subcategories) | Emissions Management, Sustainable Agriculture, Material Science, Decarbonization |

Table A6: publication count and reference mapping by AI's role in environmental problem-solving and associated environmental issue subcategories.

| Category | References | Count |
|---|---|---|
| **Forecasting** | | |
| Weather Forecasting | Charlton-Perez et al. (2024); Andersson et al. (2021); Voosen (2023a); Crespi & Voosen (2023); Voosen (2023b); Yen et al. (2019); noa (2023); Espeholt et al. (2022); Mondini et al. (2023); Schneider et al. (2023); Wong (2024); Cavaiola et al. (2024); Jose et al. (2022); Monir et al. (2023); Ebert-Uphoff & Hilburn (2023); Irrgang et al. (2021); Gibson et al. (2021); Amato et al. (2020); Buster et al. (2024); Zheng et al. (2020); Wong (2023); Thompson & Bundell (2021); Jones (2017); Ravuri et al. | 24 |
| Drought Forecasting | Al Mamun et al. (2024); Osmani et al. (2022); Devanand et al. (2024) | 3 |
| Flood Forecasting | Jones et al. (2023); noa (2024); Nearing et al. (2024); Ayyad et al. (2022); Patil et al. (2023); Jiang et al. (2024); Martinho et al. (2023) | 7 |
| Hurricane Forecasting | Balaguru et al. (2023); Ayyad et al. (2023) | 2 |
| Sea Level Rise Forecasting | Nieves et al. (2021); Ayinde et al. (2023); Bolibar et al. (2022); Tiggeloven et al. (2021) | 4 |
| Groundwater Forecasting | Wunsch et al. (2022); Sarkar et al. (2024); O et al. (2022) | 3 |
| Ocean Acidification Forecasting | Krasting et al. (2022) | 1 |
| Water Quality Forecasting | Zhi et al. (2024); Kruk et al. (2022); Zhi et al. (2023) | 3 |
| Wildfire Forecasting | Yu et al. (2022); Shadrin et al. (2024) | 2 |
| Precipitation Forecasting | Ham et al. (2023); Cornwall (2019); Mohammadi et al. (2022); Bird et al. (2023) | 4 |
| Air Quality Forecasting | Carbo-Bustinza et al. (2022); Ojha et al. (2021); Li et al. (2023); Halder et al. (2023); Gutiérrez-Avila et al. (2022) | 5 |
| Carbon Cycle Forecasting | Liu et al. (2024); Couespel et al. (2024); Joshi et al. (2024) | 3 |
| Emissions Forecasting | Jablonka et al. (2023) | 1 |
| Climate Model Forecasting | Kubečka et al. (2023); Beucler et al. (2024); Bonavita et al. (2023); Kadow et al. (2020); Mansfield et al. (2020); Mooers et al. (2023); Wang & Li (2024); Voosen (2018); Yuval & O'Gorman (2020); Hourdin et al. (2023) | 10 |
| Environmental System Forecasting | Salonen et al. (2019); Reichstein et al. (2019); Gettelman et al. (2022) | 3 |
| Volcanic Eruption and Seismic Events Forecasting | Witze (2019); Yang et al. (2022b); Bergen et al. (2019) | 3 |
| **Monitoring & Assessment** | | |
| Deforestation | Exbrayat et al. (2017); Reiner et al. (2023); Brandt et al. (2020) | 3 |
| Biodiversity | Mahecha et al. (2022); Tuia et al. (2022); Fricke et al. (2022); Thompson (2023); Petso & Jamisola (2023); Silvestro et al. (2022); Novi & Bracco (2022); Chowdhury et al. (2024); Sunkur et al. (2024); Müller et al. (2023); Pennisi (2021); O'Gorman (2022) | 12 |
| Climate Innovation | Verendel (2023) | 1 |
| Climate Systems | Callaghan et al. (2021) | 1 |
| Environmental Data Enhancement | noa (2022b) | 1 |
| Water Systems | Biermann et al. (2020) | 1 |
| Environmental Monitoring | Joppa (2017) | 1 |
| Sustainability Goals | Vinuesa et al. (2020) | 1 |
| **Mitigation** | | |
| Emissions Management | Kaack et al. (2022); noa (2022a) | 2 |
| Sustainable Agriculture | Tarek et al. (2023) | 1 |
| Material Science | Tabor et al. (2018); Yashinski (2024); Yang et al. (2022a) | 3 |
| Decarbonization | Nature Research Custom Media: Skoltech (2021) | 1 |

Table A7: Responses to Q17: "Does the publication mention whether the environmental issue was previously addressed using other non-AI approaches?"

| Response | Number of publications | Percentage |
|---|---|---|
| Yes | 86 | 81.1% |
| No | 20 | 18.9% |

Table A8: Responses to Q17 (Previous non-AI Approaches mentioned) by Environmental Issue, grouped by AI's Role in Environmental Problem-Solving

| Category | Yes | No | Total |
|---|---|---|---|
| **Forecasting** | | | |
| Weather Forecasting | 23 | 1 | 24 |
| Drought Forecasting | 3 | 0 | 3 |
| Flood Forecasting | 7 | 0 | 7 |
| Hurricane Forecasting | 2 | 0 | 2 |
| Sea Level Rise Forecasting | 4 | 0 | 4 |
| Groundwater Forecasting | 3 | 0 | 3 |
| Ocean Acidification Forecasting | 1 | 0 | 1 |
| Water Quality Forecasting | 1 | 2 | 3 |
| Wildfire Forecasting | 2 | 0 | 2 |
| Precipitation Forecasting | 4 | 0 | 4 |
| Air Quality Forecasting | 4 | 1 | 5 |
| Carbon Cycle Forecasting | 3 | 0 | 3 |
| Emissions Forecasting | 1 | 0 | 1 |
| Climate Model Forecasting | 9 | 1 | 10 |
| Environmental System Forecasting | 3 | 0 | 3 |
| Volcanic Eruption Forecasting | 3 | 0 | 3 |
| **Monitoring & Assessment** | | | |
| Deforestation | 2 | 1 | 3 |
| Biodiversity | 4 | 8 | 12 |
| Climate Innovation | 0 | 1 | 1 |
| Climate Systems | 0 | 1 | 1 |
| Environmental Data | 1 | 0 | 1 |
| Water Systems | 0 | 1 | 1 |
| Environmental Monitoring | 0 | 1 | 1 |
| Sustainability Goals | 0 | 1 | 1 |
| **Mitigation** | | | |
| Emissions Management | 2 | 0 | 2 |
| Sustainable Agriculture | 0 | 1 | 1 |
| Material Science | 3 | 0 | 3 |
| Decarbonization | 1 | 0 | 1 |

Table A9: Distribution of AI Novelty Scores in Environmental Studies (n=106)

| Novelty Score | Interpretation | Number of publications | Percentage |
|---|---|---|---|
| 1 | Highly novel/experimental | 22 | 20.75% |
| 2 | Moderately novel/established | 52 | 49.06% |
| 3 | Conventional/standard methods | 30 | 28.30% |
| NA | Not available/unclear | 2 | 1.89% |
| **Total** | | **106** | **100.00%** |

*Note*: Novelty scores were assigned on a scale from 1 (high novelty) to 3 (low novelty), based on the method's position within the machine learning research landscape and its experimental maturity.

Table A10: Average novelty score by AI role in environmental problem-solving

| AI Role | Publications Number | Percentage | Mean Novelty | Standard Deviation |
|---|---|---|---|---|
| Forecasting | 78 | 73.6% | 2.08 | 0.69 |
| Monitoring & Assessment | 21 | 19.8% | 2.16 | 0.67 |
| Mitigation | 7 | 6.6% | 1.86 | 0.83 |

APPENDIX REFERENCES

Achieving net zero emissions with machine learning: the challenge ahead. *Nature Machine Intelligence*, 4(8):661–662, 2022a. doi: 10.1038/s42256-022-00529-w.

Ways forward for Machine Learning to make useful global environmental datasets from legacy observations and measurements. *Nature Communications*, 13(1):5178, 2022b. doi: 10.1038/s41467-022-32693-3.

Deep learning shows how global warming affects daily rainfall. *Nature*, pp. d41586–023–02803–2, 2023. doi: 10.1038/d41586-023-02803-2.

Artificial intelligence can provide accurate forecasts of extreme floods at global scale. *Nature*, pp. d41586–024–00835–w, April 2024. doi: 10.1038/d41586-024-00835-w.

Md. Abdullah Al Mamun, Mou Rani Sarker, Md Abdur Rouf Sarkar, Sujit Kumar Roy, Sheikh Arafat Islam Nihad, Andrew M. McKenzie, Md. Ismail Hossain, and Md. Shahjahan Kabir. Identification of influential weather parameters and seasonal drought prediction in Bangladesh using machine learning algorithm. *Scientific Reports*, 14(1):566, January 2024. doi: 10.1038/s41598-023-51111-2.

Federico Amato, Fabian Guignard, Sylvain Robert, and Mikhail Kanevski. A novel framework for spatio-temporal prediction of environmental data using deep learning. *Scientific Reports*, 10(1): 22243, 2020. doi: 10.1038/s41598-020-79148-7.

Tom R. Andersson, J. Scott Hosking, María Pérez-Ortiz, Brooks Paige, Andrew Elliott, Chris Russell, Stephen Law, Daniel C. Jones, Jeremy Wilkinson, Tony Phillips, James Byrne, Steffen Tietsche, Beena Balan Sarojini, Eduardo Blanchard-Wrigglesworth, Yevgeny Aksenov, Rod Downie, and Emily Shuckburgh. Seasonal Arctic sea ice forecasting with probabilistic deep learning. *Nature Communications*, 12(1):5124, 2021. doi: 10.1038/s41467-021-25257-4.

Akeem Shola Ayinde, Huaming Yu, and Kejian Wu. Sea level variability and modeling in the Gulf of Guinea using supervised machine learning. *Scientific Reports*, 13(1):21318, 2023. doi: 10.1038/s41598-023-48624-1.

Mahmoud Ayyad, Muhammad R. Hajj, and Reza Marsooli. Machine learning-based assessment of storm surge in the New York metropolitan area. *Scientific Reports*, 12(1):19215, 2022. doi: 10.1038/s41598-022-23627-6.

Mahmoud Ayyad, Muhammad R. Hajj, and Reza Marsooli. Climate change impact on hurricane storm surge hazards in New York/New Jersey Coastlines using machine-learning. *npj Climate and Atmospheric Science*, 6(1):88, 2023. doi: 10.1038/s41612-023-00420-4.

Karthik Balaguru, Wenwei Xu, Chuan-Chieh Chang, L. Ruby Leung, David R. Judi, Samson M. Hagos, Michael F. Wehner, James P. Kossin, and Mingfang Ting. Increased U.S. coastal hurricane risk under climate change. *Science Advances*, 9(14):eadf0259, 2023. doi: 10.1126/sciadv.adf0259.

Karianne J. Bergen, Paul A. Johnson, Maarten V. De Hoop, and Gregory C. Beroza. Machine learning for data-driven discovery in solid Earth geoscience. *Science*, 363(6433):eaau0323, 2019. doi: 10.1126/science.aau0323.

Tom Beucler, Pierre Gentine, Janni Yuval, Ankitesh Gupta, Liran Peng, Jerry Lin, Sungduk Yu, Stephan Rasp, Fiaz Ahmed, Paul A. O'Gorman, J. David Neelin, Nicholas J. Lutsko, and Michael Pritchard. Climate-invariant machine learning. *Science Advances*, 10(6):eadj7250, 2024. doi: 10.1126/sciadv.adj7250.

Lauren Biermann, Daniel Clewley, Victor Martinez-Vicente, and Konstantinos Topouzelis. Finding Plastic Patches in Coastal Waters using Optical Satellite Data. *Scientific Reports*, 10(1):5364, 2020. doi: 10.1038/s41598-020-62298-z.

Leroy J. Bird, Gregory E. Bodeker, and Kyle R. Clem. Sensitivity of extreme precipitation to climate change inferred using artificial intelligence shows high spatial variability. *Communications Earth & Environment*, 4(1):469, 2023. doi: 10.1038/s43247-023-01142-4.

Jordi Bolibar, Antoine Rabatel, Isabelle Gouttevin, Harry Zekollari, and Clovis Galiez. Nonlinear sensitivity of glacier mass balance to future climate change unveiled by deep learning. *Nature Communications*, 13(1):409, 2022. doi: s41467-022-28033-0.

Massimo Bonavita, Rochelle Schneider, Rossella Arcucci, Matthew Chantry, Marcin Chrust, Alan Geer, Bertrand Le Saux, and Claudia Vitolo. 2022 ECMWF-ESA workshop report: current status, progress and opportunities in machine learning for Earth System observation and prediction. *npj Climate and Atmospheric Science*, 6(1):87, 2023. doi: 10.1038/s41612-023-00387-2.

Martin Brandt, Compton J. Tucker, Ankit Kariryaa, Kjeld Rasmussen, Christin Abel, Jennifer Small, Jerome Chave, Laura Vang Rasmussen, Pierre Hiernaux, Abdoul Aziz Diouf, Laurent Kergoat, Ole Mertz, Christian Igel, Fabian Gieseke, Johannes Schöning, Sizhuo Li, Katherine Melocik, Jesse Meyer, Scott Sinno, Eric Romero, Erin Glennie, Amandine Montagu, Morgane Dendoncker, and Rasmus Fensholt. An unexpectedly large count of trees in the West African Sahara and Sahel. *Nature*, 587(7832):78–82, 2020. doi: 10.1038/s41586-020-2824-5.

Grant Buster, Brandon N. Benton, Andrew Glaws, and Ryan N. King. High-resolution meteorology with climate change impacts from global climate model data using generative machine learning. *Nature Energy*, 9(7):894–906, 2024. doi: 10.1038/s41560-024-01507-9.

Max Callaghan, Carl-Friedrich Schleussner, Shruti Nath, Quentin Lejeune, Thomas R. Knutson, Markus Reichstein, Gerrit Hansen, Emily Theokritoff, Marina Andrijevic, Robert J. Brecha, Michael Hegarty, Chelsea Jones, Kaylin Lee, Agathe Lucas, Nicole Van Maanen, Inga Menke, Peter Pfleiderer, Burcu Yesil, and Jan C. Minx. Machine-learning-based evidence and attribution mapping of 100,000 climate impact studies. *Nature Climate Change*, 11(11):966–972, 2021. doi: 10.1038/s41558-021-01168-6.

Natalí Carbo-Bustinza, Marisol Belmonte, Vasti Jimenez, Paula Montalban, Magiory Rivera, Fredi Gutiérrez Martínez, Mohamed Mehdi Hadi Mohamed, Alex Rubén Huamán De La Cruz, Kleyton Da Costa, and Javier Linkolk López-Gonzales. A machine learning approach to analyse ozone concentration in metropolitan area of Lima, Peru. *Scientific Reports*, 12(1):22084, 2022. doi: 10.1038/s41598-022-26575-3.

Mattia Cavaiola, Federico Cassola, Davide Sacchetti, Francesco Ferrari, and Andrea Mazzino. Hybrid AI-enhanced lightning flash prediction in the medium-range forecast horizon. *Nature Communications*, 15(1):1188, 2024. doi: 10.1038/s41467-024-44697-2.

Andrew J. Charlton-Perez, Helen F. Dacre, Simon Driscoll, Suzanne L. Gray, Ben Harvey, Natalie J. Harvey, Kieran M. R. Hunt, Robert W. Lee, Ranjini Swaminathan, Remy Vandaele, and Ambrogio Volonté. Do AI models produce better weather forecasts than physics-based models? A quantitative evaluation case study of Storm Ciarán. *npj Climate and Atmospheric Science*, 7 (1):93, 2024. doi: 10.1038/s41612-024-00638-w.

Masuma Chowdhury, Alejo Martínez-Sansigre, Maruška Mole, Eduardo Alonso-Peleato, Nadiia Basos, Jose Manuel Blanco, Maria Ramirez-Nicolas, Isabel Caballero, and Ignacio De La Calle. AI-driven remote sensing enhances Mediterranean seagrass monitoring and conservation to combat climate change and anthropogenic impacts. *Scientific Reports*, 14(1):8360, 2024. doi: 10.1038/s41598-024-59091-7.

Warren Cornwall. Artificial intelligence could predict El Niño up to 18 months in advance. *Science*, 2019. doi: 10.1126/science.aaz5504.

Damien Couespel, Jerry Tjiputra, Klaus Johannsen, Pradeebane Vaittinada Ayar, and Bjørnar Jensen. Machine learning reveals regime shifts in future ocean carbon dioxide fluxes inter-annual variability. *Communications Earth & Environment*, 5(1):99, 2024. doi: 10.1038/s43247-024-01257-2.

Sarah Crespi and Paul Voosen. AI improves weather prediction, and cutting emissions from landfills, 2023.

Anjana Devanand, Georgina M. Falster, Zoe E. Gillett, Sanaa Hobeichi, Chiara M. Holgate, Chenhui Jin, Mengyuan Mu, Tess Parker, Sami W. Rifai, Kathleen S. Rome, Milica Stojanovic, Elisabeth Vogel, Nerilie J. Abram, Gab Abramowitz, Sloan Coats, Jason P. Evans, Ailie J. E. Gallant,

Andy J. Pitman, Scott B. Power, Surendra P. Rauniyar, Andréa S. Taschetto, and Anna M. Ukkola. Australia's Tinderbox Drought: An extreme natural event likely worsened by human-caused climate change. *Science Advances*, 10(10):eadj3460, 2024. doi: 10.1126/sciadv.adj3460.

Imme Ebert-Uphoff and Kyle Hilburn. The outlook for AI weather prediction. *Nature*, 619(7970): 473–474, 2023. doi: 10.1038/d41586-023-02084-9.

Lasse Espeholt, Shreya Agrawal, Casper Sønderby, Manoj Kumar, Jonathan Heek, Carla Bromberg, Cenk Gazen, Rob Carver, Marcin Andrychowicz, Jason Hickey, Aaron Bell, and Nal Kalchbrenner. Deep learning for twelve hour precipitation forecasts. *Nature Communications*, 13(1):5145, 2022. doi: 10.1038/s41467-022-32483-x.

Jean-François Exbrayat, Yi Y. Liu, and Mathew Williams. Impact of deforestation and climate on the Amazon Basin's above-ground biomass during 1993–2012. *Scientific Reports*, 7(1):15615, 2017. doi: 10.1038/s41598-017-15788-6.

Evan C. Fricke, Alejandro Ordonez, Haldre S. Rogers, and Jens-Christian Svenning. The effects of defaunation on plants' capacity to track climate change. *Science*, 375(6577):210–214, 2022. doi: 10.1126/science.abk3510.

Andrew Gettelman, Alan J. Geer, Richard M. Forbes, Greg R. Carmichael, Graham Feingold, Derek J. Posselt, Graeme L. Stephens, Susan C. Van Den Heever, Adam C. Varble, and Paquita Zuidema. The future of Earth system prediction: Advances in model-data fusion. *Science Advances*, 8(14):eabn3488, 2022. doi: 10.1126/sciadv.abn3488.

Peter B. Gibson, William E. Chapman, Alphan Altinok, Luca Delle Monache, Michael J. DeFlorio, and Duane E. Waliser. Training machine learning models on climate model output yields skillful interpretable seasonal precipitation forecasts. *Communications Earth & Environment*, 2(1):159, 2021. doi: 10.1038/s43247-021-00225-4.

Iván Gutiérrez-Avila, Kodi B. Arfer, Daniel Carrión, Johnathan Rush, Itai Kloog, Aaron R. Naeger, Michel Grutter, Víctor Hugo Páramo-Figueroa, Horacio Riojas-Rodríguez, and Allan C. Just. Prediction of daily mean and one-hour maximum PM2.5 concentrations and applications in Central Mexico using satellite-based machine-learning models. *Journal of Exposure Science & Environmental Epidemiology*, 32(6):917–925, 2022. doi: 10.1038/s41370-022-00471-4.

Bijay Halder, Iman Ahmadianfar, Salim Heddam, Zainab Haider Mussa, Leonardo Goliatt, Mou Leong Tan, Zulfaqar Sa'adi, Zainab Al-Khafaji, Nadhir Al-Ansari, Ali H. Jawad, and Zaher Mundher Yaseen. Machine learning-based country-level annual air pollutants exploration using Sentinel-5P and Google Earth Engine. *Scientific Reports*, 13(1):7968, 2023. doi: 10.1038/s41598-023-34774-9.

Yoo-Geun Ham, Jeong-Hwan Kim, Seung-Ki Min, Daehyun Kim, Tim Li, Axel Timmermann, and Malte F. Stuecker. Anthropogenic fingerprints in daily precipitation revealed by deep learning. *Nature*, 622(7982):301–307, 2023. doi: 10.1038/s41586-023-06474-x.

Frédéric Hourdin, Brady Ferster, Julie Deshayes, Juliette Mignot, Ionela Musat, and Daniel Williamson. Toward machine-assisted tuning avoiding the underestimation of uncertainty in climate change projections. *Science Advances*, 9(29), 2023. doi: 10.1126/sciadv.adf2758.

Christopher Irrgang, Niklas Boers, Maike Sonnewald, Elizabeth A. Barnes, Christopher Kadow, Joanna Staneva, and Jan Saynisch-Wagner. Towards neural Earth system modelling by integrating artificial intelligence in Earth system science. *Nature Machine Intelligence*, 3(8):667–674, 2021. doi: 10.1038/s42256-021-00374-3.

Kevin Maik Jablonka, Charithea Charalambous, Eva Sanchez Fernandez, Georg Wiechers, Juliana Monteiro, Peter Moser, Berend Smit, and Susana Garcia. Machine learning for industrial processes: Forecasting amine emissions from a carbon capture plant. *Science Advances*, 9(1): eadc9576, 2023. doi: 10.1126/sciadv.adc9576.

Shijie Jiang, Larisa Tarasova, Guo Yu, and Jakob Zscheischler. Compounding effects in flood drivers challenge estimates of extreme river floods. *Science Advances*, 10(13):eadl4005, 2024. doi: 10.1126/sciadv.adl4005.

Anne Jones, Julian Kuehnert, Paolo Fraccaro, Ophélie Meuriot, Tatsuya Ishikawa, Blair Edwards, Nikola Stoyanov, Sekou L. Remy, Kommy Weldemariam, and Solomon Assefa. AI for climate impacts: applications in flood risk. *npj Climate and Atmospheric Science*, 6(1):63, 2023. doi: 10.1038/s41612-023-00388-1.

Nicola Jones. How machine learning could help to improve climate forecasts. *Nature*, 548(7668): 379–379, 2017. doi: 10.1038/548379a.

Lucas N. Joppa. The case for technology investments in the environment. *Nature*, 552(7685): 325–328, 2017. doi: 10.1038/d41586-017-08675-7.

Dinu Maria Jose, Amala Mary Vincent, and Gowdagere Siddaramaiah Dwarakish. Improving multiple model ensemble predictions of daily precipitation and temperature through machine learning techniques. *Scientific Reports*, 12(1):4678, 2022. doi: 10.1038/s41598-022-08786-w.

A.P. Joshi, Prasanna Kanti Ghoshal, Kunal Chakraborty, and V. V. S. S. Sarma. Sea-surface pCO2 maps for the Bay of Bengal based on advanced machine learning algorithms. *Scientific Data*, 11 (1):384, 2024. doi: 10.1038/s41597-024-03236-w.

Lynn H. Kaack, Priya L. Donti, Emma Strubell, George Kamiya, Felix Creutzig, and David Rolnick. Aligning artificial intelligence with climate change mitigation. *Nature Climate Change*, 12(6): 518–527, 2022. doi: 10.1038/s41558-022-01377-7.

Christopher Kadow, David Matthew Hall, and Uwe Ulbrich. Artificial intelligence reconstructs missing climate information. *Nature Geoscience*, 13(6):408–413, 2020. doi: 10.1038/s41561-020-0582-5.

John P. Krasting, Maurizia De Palma, Maike Sonnewald, John P. Dunne, and Jasmin G. John. Regional sensitivity patterns of Arctic Ocean acidification revealed with machine learning. *Communications Earth & Environment*, 3(1):91, 2022. doi: 10.1038/s43247-022-00419-4.

Marek Kruk, Anna Maria Goździejewska, and Piotr Artiemjew. Predicting the effects of winter water warming in artificial lakes on zooplankton and its environment using combined machine learning models. *Scientific Reports*, 12(1):16145, 2022. doi: 10.1038/s41598-022-20604-x.

Jakub Kubečka, Yosef Knattrup, Morten Engsvang, Andreas Buchgraitz Jensen, Daniel Ayoubi, Haide Wu, Ove Christiansen, and Jonas Elm. Current and future machine learning approaches for modeling atmospheric cluster formation. *Nature Computational Science*, 3(6):495–503, 2023. doi: 10.1038/s43588-023-00435-0.

Lianfa Li, Jinfeng Wang, Meredith Franklin, Qian Yin, Jiajie Wu, Gustau Camps-Valls, Zhiping Zhu, Chengyi Wang, Yong Ge, and Markus Reichstein. Improving air quality assessment using physics-inspired deep graph learning. *npj Climate and Atmospheric Science*, 6(1):152, 2023. doi: 10.1038/s41612-023-00475-3.

Licheng Liu, Wang Zhou, Kaiyu Guan, Bin Peng, Shaoming Xu, Jinyun Tang, Qing Zhu, Jessica Till, Xiaowei Jia, Chongya Jiang, Sheng Wang, Ziqi Qin, Hui Kong, Robert Grant, Symon Mezbahuddin, Vipin Kumar, and Zhenong Jin. Knowledge-guided machine learning can improve carbon cycle quantification in agroecosystems. *Nature Communications*, 15(1):357, 2024. doi: 10.1038/s41467-023-43860-5.

Miguel D. Mahecha, Ana Bastos, Friedrich J. Bohn, Nico Eisenhauer, Hannes Feilhauer, Henrik Hartmann, Thomas Hickler, Heike Kalesse-Los, Mirco Migliavacca, Friederike E. L. Otto, Jian Peng, Johannes Quaas, Ina Tegen, Alexandra Weigelt, Manfred Wendisch, and Christian Wirth. Biodiversity loss and climate extremes — study the feedbacks. *Nature*, 612(7938):30–32, 2022. doi: 10.1038/d41586-022-04152-y.

L. A. Mansfield, P. J. Nowack, M. Kasoar, R. G. Everitt, W. J. Collins, and A. Voulgarakis. Predicting global patterns of long-term climate change from short-term simulations using machine learning. *npj Climate and Atmospheric Science*, 3(1):44, 2020. doi: 10.1038/s41612-020-00148-5.

Alfeu D. Martinho, Henrique S. Hippert, and Leonardo Goliatt. Short-term streamflow modeling using data-intelligence evolutionary machine learning models. *Scientific Reports*, 13(1):13824, 2023. doi: 10.1038/s41598-023-41113-5.

Babak Mohammadi, Mir Jafar Sadegh Safari, and Saeed Vazifehkhah. IHACRES, GR4J and MISD-based multi conceptual-machine learning approach for rainfall-runoff modeling. *Scientific Reports*, 12(1):12096, 2022. doi: 10.1038/s41598-022-16215-1.

Alessandro C. Mondini, Fausto Guzzetti, and Massimo Melillo. Deep learning forecast of rainfall-induced shallow landslides. *Nature Communications*, 14(1):2466, 2023. doi: 10.1038/s41467-023-38135-y.

Md. Moniruzzaman Monir, Md. Rokonuzzaman, Subaran Chandra Sarker, Edris Alam, Md. Kamrul Islam, and Abu Reza Md. Towfiqul Islam. Spatiotemporal analysis and predicting rainfall trends in a tropical monsoon-dominated country using MAKESENS and machine learning techniques. *Scientific Reports*, 13(1):13933, 2023. doi: 10.1038/s41598-023-41132-2.

Griffin Mooers, Mike Pritchard, Tom Beucler, Prakhar Srivastava, Harshini Mangipudi, Liran Peng, Pierre Gentine, and Stephan Mandt. Comparing storm resolving models and climates via unsupervised machine learning. *Scientific Reports*, 13(1):22365, 2023. doi: 10.1038/s41598-023-49455-w.

Jörg Müller, Oliver Mitesser, H. Martin Schaefer, Sebastian Seibold, Annika Busse, Peter Kriegel, Dominik Rabl, Rudy Gelis, Alejandro Arteaga, Juan Freile, Gabriel Augusto Leite, Tomaz Nascimento De Melo, Jack LeBien, Marconi Campos-Cerqueira, Nico Blüthgen, Constance J. Tremlett, Dennis Böttger, Heike Feldhaar, Nina Grella, Ana Falconí-López, David A. Donoso, Jerome Moriniere, and Zuzana Buřivalová. Soundscapes and deep learning enable tracking biodiversity recovery in tropical forests. *Nature Communications*, 14(1):6191, 2023. doi: 10.1038/s41467-023-41693-w.

Nature Research Custom Media: Skoltech. Harnessing AI for decarbonization. *Nature*, 2021. doi: d42473-021-00508-6.

Grey Nearing, Deborah Cohen, Vusumuzi Dube, Martin Gauch, Oren Gilon, Shaun Harrigan, Avinatan Hassidim, Daniel Klotz, Frederik Kratzert, Asher Metzger, Sella Nevo, Florian Pappenberger, Christel Prudhomme, Guy Shalev, Shlomo Shenzis, Tadele Yednkachw Tekalign, Dana Weitzner, and Yossi Matias. Global prediction of extreme floods in ungauged watersheds. *Nature*, 627(8004):559–563, 2024. doi: 10.1038/s41586-024-07145-1.

Veronica Nieves, Cristina Radin, and Gustau Camps-Valls. Predicting regional coastal sea level changes with machine learning. *Scientific Reports*, 11(1):7650, 2021. doi: 10.1038/s41598-021-87460-z.

Lyuba Novi and Annalisa Bracco. Machine learning prediction of connectivity, biodiversity and resilience in the Coral Triangle. *Communications Biology*, 5(1):1359, 2022. doi: 10.1038/s42003-022-04330-8.

Sungmin O, Rene Orth, Ulrich Weber, and Seon Ki Park. High-resolution European daily soil moisture derived with machine learning (2003–2020). *Scientific Data*, 9(1):701, 2022. doi: 10.1038/s41597-022-01785-6.

Narendra Ojha, Imran Girach, Kiran Sharma, Amit Sharma, Narendra Singh, and Sachin S. Gunthe. Exploring the potential of machine learning for simulations of urban ozone variability. *Scientific Reports*, 11(1):22513, 2021. doi: 10.1038/s41598-021-01824-z.

Shabbir Ahmed Osmani, Jong-Suk Kim, Changhyun Jun, Md. Wahiduzzaman Sumon, Jongjin Baik, and Jinwook Lee. Prediction of monthly dry days with machine learning algorithms: a case study in Northern Bangladesh. *Scientific Reports*, 12(1):19717, 2022. doi: 10.1038/s41598-022-23436-x.

Eoin J. O'Gorman. Machine learning ecological networks. *Science*, 377(6609):918–919, 2022. doi: 10.1126/science.add7563.

Kalpesh Ravindra Patil, Takeshi Doi, and Swadhin K. Behera. Predicting extreme floods and droughts in East Africa using a deep learning approach. *npj Climate and Atmospheric Science*, 6(1):108, 2023. doi: 10.1038/s41612-023-00435-x.

Elizabeth Pennisi. Artificial intelligence could help biologists classify the world's tiny creatures. *Science*, 2021. doi: 10.1126/science.abj8374.

Tinao Petso and Rodrigo S. Jamisola. Wildlife conservation using drones and artificial intelligence in Africa. *Science Robotics*, 8(85):eadm7008, 2023. doi: 10.1126/scirobotics.adm7008.

Suman Ravuri, Karel Lenc, Matthew Willson, Dmitry Kangin, Remi Lam, Piotr Mirowski, Megan Fitzsimons, Maria Athanassiadou, Sheleem Kashem, Sam Madge, Rachel Prudden, Amol Mandhane, Aidan Clark, Andrew Brock, Karen Simonyan, Raia Hadsell, Niall Robinson, Ellen Clancy, Alberto Arribas, and Shakir Mohamed. Skilful precipitation nowcasting using deep generative models of radar. 597(7878):672–677. doi: 10.1038/s41586-021-03854-z.

Markus Reichstein, Gustau Camps-Valls, Bjorn Stevens, Martin Jung, Joachim Denzler, Nuno Carvalhais, and Prabhat. Deep learning and process understanding for data-driven Earth system science. *Nature*, 566(7743):195–204, 2019. doi: 10.1038/s41586-019-0912-1.

Florian Reiner, Martin Brandt, Xiaoye Tong, David Skole, Ankit Kariryaa, Philippe Ciais, Andrew Davies, Pierre Hiernaux, Jérôme Chave, Maurice Mugabowindekwe, Christian Igel, Stefan Oehmcke, Fabian Gieseke, Sizhuo Li, Siyu Liu, Sassan Saatchi, Peter Boucher, Jenia Singh, Simon Taugourdeau, Morgane Dendoncker, Xiao-Peng Song, Ole Mertz, Compton J. Tucker, and Rasmus Fensholt. More than one quarter of Africa's tree cover is found outside areas previously classified as forest. *Nature Communications*, 14(1):2258, 2023. ISSN 2041-1723.

J. Sakari Salonen, Mikko Korpela, John W. Williams, and Miska Luoto. Machine-learning based reconstructions of primary and secondary climate variables from North American and European fossil pollen data. *Scientific Reports*, 9:15805, 2019. doi: 10.1038/s41598-019-52293-4.

Showmitra Kumar Sarkar, Rhyme Rubayet Rudra, Swapan Talukdar, Palash Chandra Das, Md. Sadmin Nur, Edris Alam, Md Kamrul Islam, and Abu Reza Md. Towfiqul Islam. Future groundwater potential mapping using machine learning algorithms and climate change scenarios in Bangladesh. *Scientific Reports*, 14(1):10328, 2024. doi: 10.1038/s41598-024-60560-2.

Tapio Schneider, Swadhin Behera, Giulio Boccaletti, Clara Deser, Kerry Emanuel, Raffaele Ferrari, L. Ruby Leung, Ning Lin, Thomas Müller, Antonio Navarra, Ousmane Ndiaye, Andrew Stuart, Joseph Tribbia, and Toshio Yamagata. Harnessing AI and computing to advance climate modelling and prediction. *Nature Climate Change*, 13(9):887–889, 2023. doi: 10.1038/s41558-023-01769-3.

Dmitrii Shadrin, Svetlana Illarionova, Fedor Gubanov, Ksenia Evteeva, Maksim Mironenko, Ivan Levchunets, Roman Belousov, and Evgeny Burnaev. Wildfire spreading prediction using multimodal data and deep neural network approach. *Scientific Reports*, 14(1):2606, 2024. doi: 10.1038/s41598-024-52821-x.

Daniele Silvestro, Stefano Goria, Thomas Sterner, and Alexandre Antonelli. Improving biodiversity protection through artificial intelligence. *Nature Sustainability*, 5(5):415–424, 2022. doi: 10.1038/s41893-022-00851-6.

Reshma Sunkur, Komali Kantamaneni, Chandradeo Bokhoree, Upaka Rathnayake, and Michael Fernando. Mangrove mapping and monitoring using remote sensing techniques towards climate change resilience. *Scientific Reports*, 14(1):6949, 2024. doi: 10.1038/s41598-024-57563-4.

Daniel P. Tabor, Loïc M. Roch, Semion K. Saikin, Christoph Kreisbeck, Dennis Sheberla, Joseph H. Montoya, Shyam Dwaraknath, Muratahan Aykol, Carlos Ortiz, Hermann Tribukait, Carlos Amador-Bedolla, Christoph J. Brabec, Benji Maruyama, Kristin A. Persson, and Alán Aspuru-Guzik. Accelerating the discovery of materials for clean energy in the era of smart automation. *Nature Reviews Materials*, 3(5):5–20, 2018. doi: 10.1038/s41578-018-0005-z.

Zahraa Tarek, Mohamed Elhoseny, Mohamemd I. Alghamdi, and Ibrahim M. EL-Hasnony. Leveraging three-tier deep learning model for environmental cleaner plants production. *Scientific Reports*, 13(1):19499, 2023. doi: 10.1038/s41598-023-43465-4.

Benjamin Thompson and Shamini Bundell. The AI that accurately predicts the chances of rain. *Nature*, pp. d41586–023–03552–y, October 2021. doi: 10.1038/d41586-021-02742-w.

Tosin Thompson. How AI can help to save endangered species. *Nature*, 623(7986):232–233, 2023. doi: 10.1038/d41586-023-03328-4.

Timothy Tiggeloven, Anaïs Couasnon, Chiem Van Straaten, Sanne Muis, and Philip J. Ward. Exploring deep learning capabilities for surge predictions in coastal areas. *Scientific Reports*, 11(1): 17224, 2021. doi: 10.1038/s41598-021-96674-0.

Devis Tuia, Benjamin Kellenberger, Sara Beery, Blair R. Costelloe, Silvia Zuffi, Benjamin Risse, Alexander Mathis, Mackenzie W. Mathis, Frank Van Langevelde, Tilo Burghardt, Roland Kays, Holger Klinck, Martin Wikelski, Iain D. Couzin, Grant Van Horn, Margaret C. Crofoot, Charles V. Stewart, and Tanya Berger-Wolf. Perspectives in machine learning for wildlife conservation. *Nature Communications*, 13(1):792, 2022. doi: 10.1038/s41467-022-27980-y.

Vilhelm Verendel. Tracking artificial intelligence in climate inventions with patent data. *Nature Climate Change*, 13(1):40–47, 2023. doi: 10.1038/s41558-022-01536-w.

Ricardo Vinuesa, Hossein Azizpour, Iolanda Leite, Madeline Balaam, Virginia Dignum, Sami Domisch, Anna Felländer, Simone Daniela Langhans, Max Tegmark, and Francesco Fuso Nerini. The role of artificial intelligence in achieving the Sustainable Development Goals. *Nature Communications*, 11(1):233, 2020. doi: 10.1038/s41467-019-14108-y.

Paul Voosen. Science insurgents plot a climate model driven by artificial intelligence. *Science*, 2018. doi: 10.1126/science.aau8974.

Paul Voosen. AI churns out lightning-fast forecasts as good as the weather agencies', 2023a.

Paul Voosen. AI is set to revolutionize weather forecasts. *Science*, 382(6672):748–749, 2023b. doi: 10.1126/science.adm9502.

Mingyu Wang and Jianping Li. Interpretable predictions of chaotic dynamical systems using dynamical system deep learning. *Scientific Reports*, 14(1):3143, 2024. doi: 10.1038/ s41598-024-53169-y.

Alexandra Witze. AI could help to predict eruptions. *Nature*, 567:156–157, 2019.

Carissa Wong. DeepMind AI accurately forecasts weather — on a desktop computer. *Nature*, pp. d41586–023–03552–y, 2023. doi: 10.1038/d41586-023-03552-y.

Carissa Wong. How AI is improving climate forecasts. *Nature*, 628(8009):710–712, 2024. doi: 10.1038/d41586-024-00780-8.

Andreas Wunsch, Tanja Liesch, and Stefan Broda. Deep learning shows declining groundwater levels in Germany until 2100 due to climate change. *Nature Communications*, 13(1):1221, 2022. doi: 10.1038/s41467-022-28770-2.

Jason Yang, Lei Tao, Jinlong He, Jeffrey R. McCutcheon, and Ying Li. Machine learning enables interpretable discovery of innovative polymers for gas separation membranes. *Science Advances*, 8(29):eabn9545, 2022a. doi: 10.1126/sciadv.abn9545.

Lei Yang, Xin Liu, Weiqiang Zhu, Liang Zhao, and Gregory C. Beroza. Toward improved urban earthquake monitoring through deep-learning-based noise suppression. *Science Advances*, 8(15): eabl3564, 2022b. doi: 10.1126/sciadv.abl3564.

Melisa Yashinski. Using machine learning and robotics to discover plastic substitutes. *Science Robotics*, 9(89):eadp7392, 2024. doi: 10.1126/scirobotics.adp7392.

Meng-Hua Yen, Ding-Wei Liu, Yi-Chia Hsin, Chu-En Lin, and Chii-Chang Chen. Application of the deep learning for the prediction of rainfall in Southern Taiwan. *Scientific Reports*, 9(1):12774, 2019. doi: 10.1038/s41598-019-49242-6.

Yan Yu, Jiafu Mao, Stan D. Wullschleger, Anping Chen, Xiaoying Shi, Yaoping Wang, Forrest M. Hoffman, Yulong Zhang, and Eric Pierce. Machine learning–based observation-constrained projections reveal elevated global socioeconomic risks from wildfire. *Nature Communications*, 13 (1):1250, 2022. doi: 10.1038/s41467-022-28853-0.

Janni Yuval and Paul A. O'Gorman. Stable machine-learning parameterization of subgrid processes for climate modeling at a range of resolutions. *Nature Communications*, 11(1):3295, 2020. doi: 10.1038/s41467-020-17142-3.

Gang Zheng, Xiaofeng Li, Rong-Hua Zhang, and Bin Liu. Purely satellite data–driven deep learning forecast of complicated tropical instability waves. *Science Advances*, 6(29):eaba1482, 2020. doi: 10.1126/sciadv.aba1482.

Wei Zhi, Christoph Klingler, Jiangtao Liu, and Li Li. Widespread deoxygenation in warming rivers. *Nature Climate Change*, 13(10):1105–1113, 2023. doi: 10.1038/s41558-023-01793-3.

Wei Zhi, Alison P. Appling, Heather E. Golden, Joel Podgorski, and Li Li. Deep learning for water quality. *Nature Water*, 2(3):228–241, 2024. doi: 10.1038/s44221-024-00202-z.

