# OpenReview forum: "Environmental AI Research Priorities: What They Reveal About Optimism and Misalignment"
_ICLR.cc/2026/Workshop/AFAA — Submitted to AFAA 2026_

### Official Review · Reviewer_MCCT · 2026-02-19
**Review: Environmental AI Research Priorities: What They Reveal About Optimism and Misalignment**

**Rating:** 3
**Confidence:** 4

**Summary:**

This paper presents a structured analysis of 106 publications from Nature and Science (2017–2024) that engage with both AI and environmental problem-solving. The authors classify each publication along 18 dimensions spanning portrayal/language, development stage, limitations, novelty, and publication type. The core finding is a substantial imbalance: 73.6% of publications focus on forecasting tasks, 19.8% on monitoring & assessment, and only 6.6% on mitigation. The authors argue this reveals a misalignment between AI research priorities (driven by computational tractability) and environmental urgency (which demands mitigation interventions). Additional findings include rising techno-optimism over time, infrequent mention of AI's carbon footprint (especially in research articles), and the observation that 81.1% of publications reference prior non-AI approaches to the same problem, suggesting AI is largely augmenting existing workflows rather than opening new problem domains. The paper frames itself as an interdisciplinary contribution combining STS, environmental science, and computer science perspectives.

**Strengths:**

- The paper asks whether AI-for-environment research is aligned with actual environmental urgency — a meaningful meta-scientific question. The forecasting-vs-mitigation imbalance (73.6% vs 6.6%) is a concrete, interpretable finding that could inform funding and editorial priorities.
- The 18-question codebook (Table 1) is well-articulated, spanning technical (novelty, development stage, AI type) and discursive (optimism, language bias) dimensions. This dual lens is more sophisticated than purely technical or purely sociological reviews.
- Figure 2 (the Sankey/alluvial diagram linking novelty → task type → environmental issue → development stage) is an effective visualization that communicates the paper's central narrative compactly.
- The non-AI referencing analysis (Section 3.5) provides a useful empirical anchor: AI is predominantly positioned as improving existing approaches rather than solving previously intractable problems.

**Weaknesses:**

- Sample representativeness remains a concern, even for a workshop paper. The title says "Environmental AI Research Priorities" but the corpus is 106 papers from two journals. For a workshop paper, this is acceptable scope, but the title and claims should be tempered. Suggest: "Environmental AI Research Priorities in High-Impact Journals" or similar scoping.
- The paper treats forecasting and mitigation as if they're in opposition, but improved forecasting often enables mitigation (e.g., flood early warning systems, crop yield predictions informing agricultural policy). A brief discussion acknowledging this dependency would strengthen the argument considerably.
- Even for a workshop paper, the novelty comparison between mitigation (n=7, mean=1.86) and forecasting (n=78, mean=2.08) should not be presented as a meaningful finding without acknowledging the extreme sample size imbalance. A simple caveat sentence would suffice.
-The workshop specifically focuses on agentic AI systems. The paper could note that the forecasting dominance reflects a pre-agentic paradigm, and ask what happens as environmental AI becomes more agentic (autonomous monitoring, automated mitigation decisions). This would be a natural and valuable extension.


Questions for authors
How would you connect your findings to algorithmic fairness specifically? The forecasting-mitigation imbalance could be framed as a distributional fairness issue (whose risks get predicted vs. whose emissions get mitigated), but this link isn't made explicit.
Do you see agentic environmental AI systems changing the patterns you observe? The workshop's focus on agents that "adapt and act" seems directly relevant to your mitigation gap.
Would you consider reframing the title to scope it to high-impact journals? This would preempt the most obvious critique while preserving all your findings.
Have you considered what an "aligned" distribution of AI-environment research would look like? Even a brief normative sketch would strengthen the "misalignment" claim.

---

### Official Review · Reviewer_i9rq · 2026-02-19
**Analyzing 106 Nature and Science publications from 2017-2024**

**Rating:** 3
**Confidence:** 4

**Summary:**

This paper analyzes 106 publications from Nature and Science (2017–2024) to characterize how AI is portrayed and deployed in environmental research. Using an 18-question analytical framework spanning five dimensions (portrayal/language, research focus/development stage, limitations/feasibility, AI systems/novelty, and publication type), the authors find a substantial imbalance: 73.6% of publications focus on forecasting, 19.8% on monitoring & assessment, and only 6.6% on mitigation. They document a persistent techno-optimistic bias, note that 81.1% of publications reference prior non-AI approaches (suggesting AI is largely improving existing workflows rather than solving new problems), find that most AI systems remain at research/demonstration stages with limited real-world deployment, and observe that carbon footprint considerations are rarely discussed (only ~4% of research articles). The authors frame these findings as evidence of a misalignment between AI research priorities and environmental urgency.

**Strengths:**

The paper asks whether the AI-for-environment research agenda is actually aligned with environmental priorities or is instead driven by computational tractability. This is a valuable meta-scientific question that speaks directly to the workshop's theme of alignment and fairness, specifically, whether AI development is aligned with societal needs or with what's technically convenient. The 18-question framework covering five dimensions is thoughtful and well-operationalized. It goes beyond simple bibliometric counting to capture discursive and epistemic features, optimism, limitation acknowledgment, carbon footprint discussion, novelty, and development stage. The combination of technical and STS (Science and Technology Studies) perspectives is a genuine interdisciplinary contribution. The distinction between optimism about AI in general (Q2) versus optimism about one's own model (Q3), and the finding that authors tend to be more optimistic about their own systems, is a perceptive observation about how incentive structures shape scientific discourse. Finally, the 11-to-1 ratio between forecasting and mitigation uses is a compelling headline finding. The supporting observation that mitigation tasks show higher novelty scores (1.86 vs. 2.08) but remain at early development stages enriches the interpretation: AI is being applied more experimentally in mitigation precisely because the modelling frameworks are less established.

**Weaknesses:**

I find that the sample is too small and too narrow to support the claims made. 106 publications from only two journals is a very limited corpus. Nature and Science are prestigious generalist outlets that heavily select for novelty and broad appeal; they are not representative of the environmental AI literature. The heavy forecasting skew could simply reflect these journals' preference for flashy ML-beats-physics stories (weather prediction being the prime example). The authors acknowledge this in the discussion, but don't sufficiently temper their claims throughout the paper. Statements like "a mismatch between AI research priorities and environmental urgency" overreach from what the data can support; it's more accurately a mismatch in how Nature and Science cover environmental AI.

Each publication was reviewed by a single coder, with only a 20% cross-check and no formal inter-rater reliability statistics. For inherently subjective variables like optimism level (on a 1–5 scale), novelty (1–3 scale), and whether language is "emotionally charged," this is a significant methodological weakness. The authors themselves acknowledge this ("we treat these variables as interpretive judgments"), but this substantially limits the strength of any quantitative conclusions drawn from these scores.

The paper's central normative claim is that more AI research should target mitigation. But this assumes a direct mapping between environmental urgency and where AI research should be concentrated. Forecasting is not merely academic; accurate weather and climate prediction directly supports adaptation, disaster preparedness, and policy decisions. The paper doesn't engage with the argument that better prediction is a form of environmental action, or that AI may simply not be the right tool for many mitigation challenges (which are often political, economic, or behavioural rather than computational).

Finally, I think the temporal analysis is confounded by publication volume. Trends over 2017–2024 in a corpus of 106 papers mean that individual years have very few data points (some years appear to have fewer than 5 publications). Concluding temporal shifts in optimism, focus, or novelty from such small yearly counts is statistically fragile. The 2022–2023 "peak in optimism" could easily be noise.

---

### Official Review · Reviewer_2rAZ · 2026-02-20
**Bibliometric Analysis of Environmental AI: Out of Scope for AFAA**

**Rating:** 2
**Confidence:** 4

**Summary:**

The paper analyzes 106 environmental research publications from Nature and Science (2017–2024) to characterize how AI is being used and portrayed in environmental science. It reports trends such as the dominance of forecasting applications and increasing optimism about AI’s potential. The work is primarily a descriptive meta-analysis of research priorities and discourse.

**Strengths:**

1. Provides a structured overview of AI usage trends in environmental research.
2. Uses a predefined set of categories to systematically analyze the papers.
3. Raises potentially useful discussion about research prioritization in environmental AI.

**Weaknesses:**

1. Poor fit for the AFAA workshop: fairness, alignment procedures, and agentic system behavior are not central to the study.
2. No fairness metrics, bias analysis, or mitigation methods are proposed.
3. Contribution is primarily descriptive bibliometric analysis with limited technical novelty.
4. Dataset scope (Nature/Science only) further limits generalizability.

---

### Meta-Review · Area_Chair_jY45 · 2026-02-21

**Recommendation:** Reject
**Confidence:** 4

**Metareview:**

The reviewers agree that the paper is not really a good fit for the workshop. A shared concern is that the number of sampled studies is small especially when distributed over the years, and the targeted venues of the study might not be representative of the field. Moreover, the central claim seems to be that more work should be focusing on mitigation might not be well founded, as for such applications, as also pointed by the reviewers, forecasting is the main enabler for mitigation techniques. Ultimately, I think the paper is not a good fit for the workshop and it is not ready for sharing with the community in its current form.

---

### Decision · Program_Chairs · 2026-03-02

Reject